# Machine learning models to predict traumatic brain injury outcomes in Tanzania: Using delays to emergency care as predictors

Armand Zimmerman[1], Cyrus Elahi[1], Thiago Augusto Hernandes Rocha[2], Francis Sakita[3], Blandina T. Mmbaga[3,4], Catherine A. Staton[1,5], Joao Ricardo Nickenig Vissoci[1,5]*

1 Duke Global Health Institute, Duke University, Durham, North Carolina, United States of America, 2 Federal University of Minas Gerais, Belo Horizonte, Brazil, 3 Kilimanjaro Christian Medical Centre, Moshi, Tanzania, 4 Kilimanjaro Clinical Research Institute, Moshi, Tanzania, 5 Division of Emergency Medicine, Department of Surgery, Duke University School of Medicine, Durham, North Carolina, United States of America

* joao.vissoci@duke.edu

**Data Availability Statement:** Per our study protocol that was approved by Tanzania's National Institute for Medical Research, a Data Transfer

## Abstract

Constraints to emergency department resources may prevent the timely provision of care following a patient's arrival to the hospital. In-hospital delays may adversely affect health outcomes, particularly among trauma patients who require prompt management. Prognostic models can help optimize resource allocation thereby reducing in-hospital delays and improving trauma outcomes. The objective of this study was to investigate the predictive value of delays to emergency care in machine learning based traumatic brain injury (TBI) prognostic models. Our data source was a TBI registry from Kilimanjaro Christian Medical Centre Emergency Department in Moshi, Tanzania. We created twelve unique variables representing delays to emergency care and included them in eight different machine learning based TBI prognostic models that predict in-hospital outcome. Model performance was compared using the area under the receiver operating characteristic curve (AUC). Inclusion of our twelve time to care variables improved predictability in each of our eight prognostic models. Our Bayesian generalized linear model produced the largest AUC, with a value of 89.5 (95% CI: 88.8, 90.3). Time to care variables were among the most important predictors of in-hospital outcome in our best three performing models. In low-resource settings where delays to care are highly prevalent and contribute to high mortality rates, incorporation of care delays into prediction models that support clinical decision making may benefit both emergency medicine physicians and trauma patients by improving prognostication performance.

## Introduction

Traumatic brain injury (TBI) is a leading cause of trauma related death and disability worldwide, affecting an estimated 69 million people annually [1]. The global burden of TBI is disproportionately endured by low- and middle-income countries (LMICs) which account for 72.0%

Agreement must be completed between the study investigators and the individual or entity requesting access to the study data before the data may be shared. Requests for data may be made to The National Institute for the National Institute for Medical Research at Medical Research of Tanzania at the email: (hq@nimr.or.tz).

**Funding:** Dr. Staton would like to acknowledge salary support funding from the Fogarty International Center (Staton, K01 TW010000-01A1). The funders had no role in study design, data collection and analysis, decision to publish, or preparation of the manuscript.

**Competing interests:** The authors declare that no conflicts of interest exist.

of all TBI cases and over 90.0% of all trauma related deaths [1, 2]. Among LMICs, estimates suggest the incidence of TBI is largest in sub-Saharan Africa [3]. Moreover, in comparison to high-income countries (HICs), TBI patients in the sub-Saharan region experience higher rates of mortality [4, 5].

High mortality rates among TBI patients in sub-Saharan LMICs may be partially explained using the three-delay model. The three-delay model was originally developed as a framework for assessing high maternal mortality rates in low-income settings [6]. More recently, the model has been used to understand poor outcomes among emergency care recipients [7]. The model focuses on three main causes of poor emergency patient outcomes: delays in deciding to seek care, delays in reaching a health facility, and delays in receiving adequate and appropriate treatment [7].

In the sub-Saharan region, all three delays affect trauma patients. Delays in seeking care are largely dependent on an individual's perceived need for care and their ability to access effective and affordable treatment, both of which can be influenced by cultural, geographical, and socio-economic factors. Delays in reaching a health facility can be reduced through the use of national prehospital emergency care systems. However, less than 9.0% of Africa's population has access to prehospital emergency care services [8]. Lastly, delays in receiving appropriate treatment depend on an emergency department's (ED's) availability of medical professionals and equipment. In sub-Saharan LMICs, in-hospital treatment delays have been associated with increased TBI patient mortality [9–12]. ED healthcare providers can play a critical role in reducing all three delays through research, policy, and advocacy.

From the perspective of an ED healthcare provider, and in a setting of limited resources, the first step in reducing treatment delays is optimizing the allocation of resources to patients. Prognostic models are an innovative solution that can help optimize clinical decision making to maximize positive patient outcomes. In the field of medicine, a prognostic model uses patient information to estimate the probability of a clinical outcome thereby supporting clinical decision making [13–16]. A TBI specific prognostic model would allow point of care healthcare providers to readily assess a TBI patient's prognosis at any point during the patient's hospital stay, using the patient's most up to date clinical information. Consequently, in health systems lacking trained and specialized professionals, a prognostic model could bridge gaps in clinical decision making.

Our team previously constructed and internally validated multiple machine learning based TBI prognostic models using patient data from a tertiary referral hospital in Tanzania [17]. However, these prognostic models do not include time delays that represent the three-delay model described above. The objective of this study was to create variables representing patient delays to emergency care, incorporate these variables as predictors in our baseline prognostic models, and assess how the inclusion of these predictors impact the performance of each model. To our knowledge, no TBI prognostic model to date has used delays to emergency care as a predictor of patient prognosis.

## Methods

### Study design and setting

This study is an analysis of clinical data from Kilimanjaro Christian Medical Centre (KCMC) in Moshi, Tanzania. KCMC is a tertiary referral hospital serving a population of over 15 million people [18]. Approximately 1,000 TBI patients present to the KCMC ED each year, one third of whom are admitted to the intensive care unit (ICU) [12].

   

## Study participants

This study uses a registry of 3,209 TBI patients admitted to the KCMC ED. The registry was collected prospectively from 2013 to 2017 and includes information on demographics, vital signs, injury characteristics, time to care, care received, and outcomes [12]. Patients were included in the registry if they presented with acute TBI (less than 24 hours since injury occurrence) of any severity and were evaluated by a physician in the ED. TBI patients who did not survive long enough to be evaluated by a physician in the ED, who presented for follow-up, or who presented with non-acute TBI were not included in the registry.

## Study variables

Each of the models assessed in this study included the following variables as predictors: age, sex, mechanism of injury, intention of injury, day of injury alcohol use, temperature, respiratory rate, heart rate, systolic and diastolic blood pressure, pulse oxygen, pupil reactivity, Glasgow Coma Score (GCS), ED disposition, and twelve time to care variables.

The twelve time to care variables included time from (1) injury occurrence to hospital arrival, (2) hospital arrival to physician arrival, (3) physician arrival to lab tests ordered, (4) physician arrival to chest x-ray, (5) physician arrival to skull x-ray, (6) physician arrival to brain CT scan, (7) physician arrival to administration of fluids, (8) physician arrival to administration of oxygen, (9) physician arrival to surgeon arrival, (10) physician arrival to TBI surgery, (11) physician arrival to non-TBI surgery, and (12) physician arrival to intensive care unit (ICU) admission. Time stamps for each event were recorded and entered into the TBI registry upon patient encounter/event occurrence. Our time to care variables were then calculated in the background of our registry as the difference between time stamps for different events.

The outcome predicted by our models was a patient's Glasgow Outcome Score (GOS) dichotomized as good (4–5) or poor (1–3). The Glasgow Outcome Scale is a validated measure used to assess recovery among trauma and head injury patients [19]. The score ranges from one to five with the following categories and is typically assigned at hospital discharge: (1) death, (2) persistent vegetative state, (3) severe disability, (4) moderate disability, and (5) low disability. Low disability is defined as a return to normal life with no more than minor cognitive deficits. GOS was calculated at hospital discharge for all patients, except those who died during hospitalization. We treated GOS as dichotomous, rather than continuous, because few patients in our sample had moderate GOS scores.

## Categories for time to care variables

The South African Triage Scale (SATS) is a tool used to assist patient triage in low-income EDs, and which has been validated in clinical settings across numerous LMICs [20–27]. The SATS defines standards for the timely care of patients presenting to EDs. Using these standards as a reference, we considered the following categories for our twelve time to care variables: 1.0 hours or less, 1.1–4.0 hours, 4.1–12.0 hours, and 12.1 hours or greater.

## Additional categories for time to care variables

Logically, patients in our registry who did not receive a certain procedure (e.g. x-ray, CT scan, oxygen, surgery, etc.) also did not have a recorded delay to receiving that procedure. For all such patients in our registry, we attempted to evaluate their need for any procedure they did not receive. Our evaluation of a patient's need for a procedure was based on their vital sign information upon hospital presentation. We dichotomized patient need for a procedure as "needed" or "not needed."

Needing fluids was defined as being hypotensive (<100 mmHg systolic blood pressure). Needing oxygen was defined as being hypoxic (<92% pulse oxygen) or having a GCS of 8 or less. Needing a brain CT scan was defined as having a GCS of 13 or less [28–31]. Our registry lacked sufficient data to evaluate a patient's need for any other procedures. For example, with the data available in our registry we had no way of evaluating a patient's need for surgery or ICU admission.

Our evaluation of patient need allowed us to add two additional categories to three variables: time to fluids, time to oxygen, and time to brain CT scan. These two additional categories were "patient needed but did not receive the procedure" and "patient did not need and did not receive the procedure." We also added one additional category to three other variables: time to TBI surgery, time to non-TBI surgery, and time to ICU admission. This additional category was "patient did not receive the procedure."

## Data pre-processing

All data processing was performed using the statistical software R. Variables with more than 20% of observations missing were removed from our analysis. Ten iterations of multiple imputation by chained equations, using the mice package in R software, were used to impute missing data for all remaining variables [32]. After imputation and conversion to indicator variables, highly correlated variables (correlation coefficient $\geq 0.9$ or $\leq -0.9$) as well as those with near-zero variance were removed. Our final analysis included 48 predictors and 3,140 patients.

## Predictive modeling

We produced eight different predictive models using eight different machine learning algorithms: Artificial Neural Network, Bagged Tree, Bayesian Generalized Linear Model, Gradient Boosting Machine, K-Nearest Neighbor, Random Forest, Ridge Regression, and Single C5.0 Ruleset. To train and internally validate each model, we used repeated cross validation with five repetitions of ten-fold partitioning. The measure used to define the best performing model was the area under the receiver operating characteristic (ROC) curve (AUC). However, we also report accuracy, sensitivity, specificity, positive predictive value (PPV), and negative predictive value (NPV) using a standard classification threshold of 0.50 for all models. To compare predictor importance across each of our eight models, we used the *varImp* function in R software's caret package [33]. The *varImp* function measures variable importance differently for each model type. However, to allow comparison between model types, the variable importance measures are scaled to have a minimum and maximum value of 0 and 100, respectively. Thus, variables with values of 100 are most important to a model's prediction ability while variables with values of 0 are not at all important.

## Ethics statement

This study was approved by Duke University's institutional review board, Tanzania's National Institute of Medical Research, and KCMC's ethics committee. Participant consent was waived as this study is a secondary analysis of a de-identified clinical registry.

## Results

### Sociodemographic, injury, and clinical characteristics

Overall, 2,786 (88.7%) patients experienced a good outcome and 354 (11.3%) patients experienced a poor outcome. The mean age of our sample was 30.8 years and 34.2 years for the good

and poor outcome groups, respectively. Patients in the good and poor outcome groups were predominately male (82.0%, 84.2%) and most sustained injury through a motorcycle related road traffic incident (50.1%, 50.0%). A minority of patients in the good and poor outcome groups reported day of injury alcohol use (25.7%, 23.7%). From a clinical perspective, our outcome groups differed significantly with regard to one vital sign. Patients in the good outcome group had a mean pulse oxygen of 96.4. In contrast, patients in the poor outcome group had a mean pulse oxygen of 89.1. The mean GCS score in the good and poor outcome groups was 13.9 and 8.1, respectively (Table 1).

### Time to care characteristics

In both the good and poor outcome groups the most common time to ED arrival was more than 12.0 hours after injury occurrence (31.6%, 33.3%). The most common wait time from ED arrival to physician evaluation for both the good and poor outcome groups was 0.0 to 1.0 hours (94.5%, 96.9%). Following physician evaluation, a majority of good and poor outcome patients waited 0.0 to 1.0 hours for lab tests (47.1%, 60.7%), 0.0 to 1.0 hours for a chest x-ray (48.4%, 51.4%), and 0.0 to 1.0 hours for a skull x-ray (48.6%, 54.5%). Most good outcome patients did not receive and did not need a brain CT scan (78.2%) whereas most poor outcome patients needed, but did not receive a brain CT scan (71.2%). A majority of good and poor outcome patients did not receive and did not need fluids (64.5%, 44.9%), and did not receive and did not need oxygen (94.4%, 54.8%). Most patients in both the good and poor outcome groups did not receive a TBI surgery (82.9%, 78.0%), a non-TBI surgery (90.6%, 92.1%), or ICU admission (93.9%, 70.1%) (Table 2).

### Model performance

We constructed eight different machine learning based prognostic models. The AUC of each model is depicted in Fig 1. Our Bayesian generalized linear model produced the largest AUC, with a value of 89.5 (95% CI: 88.8, 90.3). Our Bayesian model had an accuracy of 0.871, a sensitivity of 0.892, a specificity of 0.748, a PPV of 0.955, and an NPV of 0.533.

We compared the AUC of our eight models to the AUC of our eight models without our time to care to variables. For each of our eight models, the AUC increased with the inclusion of our time to care variables (Table 3). The largest increase in AUC occurred in our k nearest neighbors model, which gained 5.1 percentage points. The lowest increase in AUC occurred in our single c5.0 ruleset model, which gained 0.8 percentage points. Our best performing model, the Bayesian generalized linear model, gained 3.0 percentage points in AUC with the inclusion of our time to care variables.

We also compared the accuracy, sensitivity, specificity, PPV, and NPV of our eight models with and without the time to care variables (Table 4). Excepting the K-nearest neighbors model, most performance metrics increased with the addition of time to care variables. However, gains were minimal and often no more than three percentage points.

Lastly, we compared predictor importance across our top three performing models (Fig 2). In our Bayesian generalized linear model and ridge regression model, eight of the top twelve predictors were time to care variables. In our third best performing model, three of the top five predictors were time to care variables. Across these models, time to brain CT scan, time to oxygen, time to ICU admission, and time to fluids appear to be the most important time to care variables.

## Discussion

To our knowledge, our prediction models are the first to incorporate time to emergency care as a predictor of TBI patient outcomes in a low-income setting. Our time to care variables

**Table 1. Sociodemographic, injury, and clinical characteristics of study participants.**

| Variable | Total | Good Outcome | Poor Outcome | p-Value |
|---|---|---|---|---|
| Age, Mean (SD) | 31.2 (15.3) | 30.8 (15.2) | 34.2 (15.4) | <0.001 |
| Sex, N (%) | | | | 0.352 |
| Male | 2,583 (82.3) | 2,285 (82.0) | 298 (84.2) | |
| Female | 557 (17.7) | 501 (18.0) | 56 (15.8) | |
| Mechanism of Injury, N (%) | | | | <0.001 |
| Car RTI | 588 (18.7) | 505 (18.1) | 83 (23.4) | |
| Motorcycle RTI | 1,574 (50.1) | 1,397 (50.1) | 177 (50.0) | |
| Pedestrian RTI | 432 (13.8) | 369 (13.2) | 63 (17.8) | |
| Assault | 118 (3.8) | 111 (4.0) | 7 (2.0) | |
| Gun | 17 (0.5) | 14 (0.5) | 3 (0.8) | |
| Knife | 150 (4.8) | 143 (5.1) | 7 (2.0) | |
| Domestic Violence | 261 (8.3) | 247 (8.9) | 14 (4.0) | |
| Intention of Injury, N (%) | | | | <0.001 |
| Self-Inflicted | 2 (0.0) | 1 (0.0) | 1 (0.3) | |
| Inflicted by Other | 533 (17.0) | 500 (17.9) | 33 (9.3) | |
| Unintentional | 2,585 (82.3) | 2,272 (81.6) | 313 (88.4) | |
| Unknown | 20 (0.6) | 13 (0.5) | 7 (2.0) | |
| Day of Injury Alcohol Use, N (%) | | | | <0.001 |
| Yes | 799 (25.4) | 715 (25.7) | 84 (23.7) | |
| No | 1,533 (48.8) | 1,399 (50.2) | 134 (37.9) | |
| Unknown | 808 (25.7) | 672 (24.1) | 136 (38.4) | |
| Temperature, Mean (SD) | 36.5 (0.7) | 36.4 (0.7) | 36.5 (1.1) | 0.09 |
| Respiratory Rate, Mean (SD) | 21.8 (3.9) | 21.6 (3.7) | 22.8 (5.2) | <0.001 |
| Heart Rate, Mean (SD) | 88.0 (18.2) | 87.6 (17.4) | 91.3 (23.6) | 0.004 |
| Systolic Blood Pressure, Mean (SD) | 121.7 (20.2) | 121.9 (19.4) | 120.5 (25.5) | 0.333 |
| Diastolic Blood Pressure, Mean (SD) | 72.6 (14.8) | 72.8 (14.4) | 71.2 (17.6) | 0.092 |
| Pulse Oxygen, Mean (SD) | 95.5 (7.2) | 96.4 (5.2) | 89.1 (14.1) | <0.001 |
| GCS, Mean (SD) | 13.3 (3.2) | 13.9 (2.4) | 8.1 (4.3) | <0.001 |
| Right Pupil, N (%) | | | | <0.001 |
| Reactive | 2,992 (95.3) | 2,747 (98.6) | 245 (69.2) | |
| Not Reactive | 135 (4.3) | 29 (1.0) | 106 (29.9) | |
| Unknown | 13 (0.4) | 10 (0.4) | 3 (0.8) | |
| Left Pupil, N (%) | | | | <0.001 |
| Reactive | 3,017 (96.1) | 2,752 (98.8) | 265 (74.9) | |
| Not Reactive | 109 (3.5) | 23 (0.8) | 86 (24.3) | |
| Unknown | 14 (0.4) | 11 (0.4) | 3 (0.8) | |
| Pupils Equal, N (%) | | | | <0.001 |
| Yes | 908 (28.9) | 824 (29.6) | 84 (23.7) | |
| No | 20 (0.6) | 8 (0.3) | 12 (3.4) | |
| Unknown | 2,212 (70.4) | 1,954 (70.1) | 258 (72.9) | |
| ED Disposition, N (%) | | | | <0.001 |
| ICU | 86 (2.7) | 34 (5.4) | 52 (14.6) | |
| Surgery | 2,674 (85.2) | 239 (38.2) | 276 (77.7) | |
| Operating Theatre | 14 (0.4) | 13 (2.1) | 1 (0.3) | |
| Home | 340 (10.8) | 340 (54.3) | 0 (0.0) | |
| Death | 26 (0.8) | 0 (0.0) | 26 (7.3) | |

**Table 2. Time to care characteristics of study participants.**

| Variable | Total | Good Outcome | Poor Outcome | p-Value |
|---|---|---|---|---|
| **Time to Arrival, N (%)** | | | | 0.791 |
| 0.0–1.0 h | 592 (18.9) | 522 (18.7) | 70 (19.8) | |
| 1.1–4.0 h | 975 (31.1) | 869 (31.2) | 106 (29.9) | |
| 4.1–12.0 h | 575 (18.3) | 515 (18.5) | 60 (16.9) | |
| >12.0 h | 998 (31.8) | 880 (31.6) | 118 (33.3) | |
| **Time to ED Physician, N (%)** | | | | 0.046 |
| 0.0–1.0 h | 2,976 (94.8) | 2,633 (94.5) | 343 (96.9) | |
| 1.1–4.0 h | 118 (3.8) | 113 (4.1) | 5 (1.4) | |
| >4.0 h | 46 (1.5) | 40 (1.4) | 6 (1.7) | |
| **Time to Lab Tests Ordered, N (%)** | | | | <0.001 |
| 0.0–1.0 h | 1,526 (48.6) | 1,311 (47.1) | 215 (60.7) | |
| 1.1–4.0 h | 1,069 (34.0) | 975 (35.0) | 94 (26.6) | |
| 4.1–12.0 h | 342 (10.9) | 318 (11.4) | 24 (6.8) | |
| >12.0 h | 203 (6.5) | 182 (6.5) | 21 (5.9) | |
| **Time to Chest X-ray, N (%)** | | | | 0.027 |
| 0.0–1.0 h | 1,531 (48.8) | 1,349 (48.4) | 182 (51.4) | |
| 1.1–4.0 h | 1,047 (33.3) | 950 (34.1) | 97 (27.4) | |
| >4.0 h | 562 (17.9) | 487 (17.5) | 75 (21.2) | |
| **Time to Skull X-ray, N (%)** | | | | 0.008 |
| 0.0–1.0 h | 1,546 (49.2) | 1,353 (48.6) | 193 (54.5) | |
| 1.1–4.0 h | 1,210 (38.5) | 1,100 (39.5) | 110 (31.1) | |
| >4.0 h | 384 (12.2) | 333 (12.0) | 51 (14.4) | |
| **Time to Brain CT Scan, N (%)** | | | | <0.001 |
| 0.0–1.0 h | 37 (1.2) | 29 (1.0) | 8 (2.3) | |
| 1.1–4.0 h | 54 (1.7) | 45 (1.6) | 9 (2.5) | |
| >4.0 h | 50 (1.6) | 45 (1.6) | 5 (1.4) | |
| Not Received, Needed | 741 (23.6) | 489 (17.6) | 252 (71.2) | |
| Not Received, Not Needed | 2,258 (71.9) | 2,178 (78.2) | 80 (22.6) | |
| **Time to Fluid Administration, N (%)** | | | | <0.001 |
| 0.0–1.0 h | 812 (25.9) | 672 (24.1) | 140 (39.5) | |
| 1.1–4.0 h | 102 (3.2) | 88 (3.2) | 14 (4.0) | |
| >4.0 h | 126 (4.0) | 111 (4.0) | 15 (4.2) | |
| Not Received, Needed | 144 (4.6) | 118 (4.2) | 26 (7.3) | |
| Not Received, Not Needed | 1,956 (62.3) | 1,797 (64.5) | 159 (44.9) | |
| **Time to Oxygen Administration, N (%)** | | | | <0.001 |
| 0.0–1.0 h | 77 (2.5) | 25 (0.9) | 52 (14.7) | |
| >1.0 h | 20 (0.6) | 5 (0.2) | 15 (4.2) | |
| Not Received, Needed | 219 (7.0) | 126 (4.5) | 93 (26.3) | |
| Not Received, Not Needed | 2,824 (90.0) | 2,630 (94.4) | 194 (54.8) | |
| **Time to TBI Surgery, N (%)** | | | | 0.027 |
| 0.0–4.0 h | 77 (2.5) | 71 (2.5) | 6 (1.7) | |
| 4.1–12.0 h | 165 (5.3) | 142 (5.1) | 23 (6.5) | |
| >12.0 h | 312 (9.9) | 263 (9.4) | 49 (13.8) | |
| Not Received | 2,586 (82.4) | 2,310 (82.9) | 276 (78.0) | |
| **Time to Other Surgery, N (%)** | | | | 0.186 |
| 0.0–4.0 h | 51 (1.6) | 42 (1.5) | 9 (2.5) | |
| 4.1–12.0 h | 83 (2.6) | 76 (2.7) | 7 (2.0) | |

*(Continued)*

**Table 2.** (Continued)

| Variable | Total | Good Outcome | Poor Outcome | p-Value |
|---|---|---|---|---|
| >12.0 h | 156 (5.0) | 144 (5.2) | 12 (3.4) | |
| Not Received | 2,850 (90.8) | 2,524 (90.6) | 326 (92.1) | |
| **Time to ICU Admission, N (%)** | | | | <0.001 |
| 0.0–1.0 h | 14 (0.4) | 5 (0.2) | 9 (2.5) | |
| 1.1–4.0 h | 50 (1.6) | 19 (0.7) | 31 (8.8) | |
| >4.0 h | 212 (6.8) | 146 (5.2) | 66 (18.6) | |
| Not Received | 2,864 (91.2) | 2,616 (93.9) | 248 (70.1) | |

encompass both receipt of hospital care (whether or not a patient received a specific diagnostic or treatment procedure) and delays to hospital care (how long a patient waited before receiving a specific diagnostic or treatment procedure). The value of time to care variables lie not only in

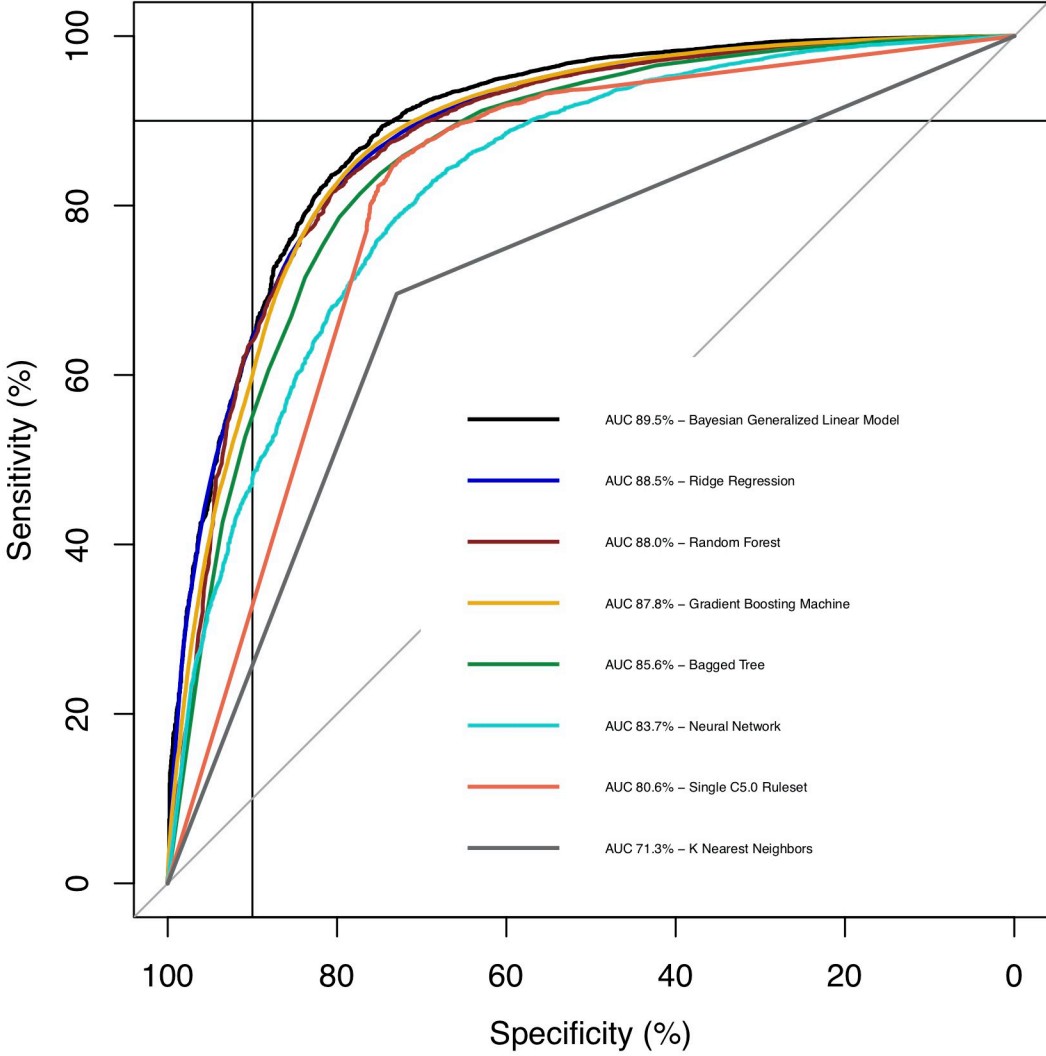

**Fig 1. Comparison of receiver operating characteristic (ROC) curves across all models after the inclusion of time to care variables.** The Bayesian generalized linear model had the largest area under the ROC curve (AUC).

**Table 3. AUC of each model with and without time to care variables.**

| Prediction Algorithm | AUC (95% CI)—Without Time to Care Variables | AUC (95% CI)—With Time to Care Variables | Difference in AUC |
|---|---|---|---|
| K Nearest Neighbors | 66.2 (66.1, 65.5) | 71.3 (71.1, 71.5) | 5.1 |
| Neural Network | 78.8 (77.8, 80.0) | 83.7 (82.8, 84.7) | 4.9 |
| Ridge Regression | 84.8 (84.5, 85.3) | 88.5 (88.3, 88.8) | 3.7 |
| Random Forest | 84.9 (84.6, 85.3) | 88.0 (87.2, 88.9) | 3.1 |
| Bayesian Generalized Linear Model | 86.5 (85.6, 87.4) | 89.5 (88.8, 90.3) | 3.0 |
| Gradient Boosting Machine | 85.1 (84.9, 85.3) | 87.8 (87.7, 88.0) | 2.7 |
| Bagged Tree | 83.6 (82.7, 84.6) | 85.6 (84.7, 86.5) | 2.0 |
| Single C5.0 Ruleset | 79.8 (78.8, 80.9) | 80.6 (79.6, 81.6) | 0.8 |

their potential to improve model performance, but also in their potential to improve translation of machine learning based prediction models into clinical decision support tools. Based on the results presented in this study, we suggest the following: inclusion of time to care predictors (1) increases the clinical relevance of TBI prediction models, (2) improves the usefulness of TBI prediction models as clinical decision support tools, and (3) improves the usability of TBI prediction models as clinical decision support tools.

## Clinical relevance and time to care

In accordance with the Prognosis Research Series on prognostic model research, we updated our prediction models to include twelve new time to care variables as predictors [34]. In addition, we compared the performance of our baseline and updated models using discrimination metrics [35]. Our results provide two important points regarding the clinical relevance of time to care in TBI prediction models.

First, time to care variables improve model performance. Inclusion of our twelve time to care variables improved prediction performance in each of our models as indicated by an increase in AUC (Table 3) and other performance metrics (Table 4). Our updated models not only outperformed our baseline models, but also performed similarly to other published TBI prediction models. In a systematic review of 102 TBI prediction models, the highest reported AUC was 89.0 [36]. In a similar review, only two of eleven identified TBI prediction models reported AUC's greater than 80.0 [37]. An additional review of neurosurgical prediction

**Table 4. Performance metrics of each model with and without time to care variables.**

| Prediction Algorithm | Accuracy (Without) | Accuracy (With) | Sensitivity (Without) | Sensitivity (With) | Specificity (Without) | Specificity (With) | PPV (Without) | PPV (With) | NPV (Without) | NPV (With) |
|---|---|---|---|---|---|---|---|---|---|---|
| K Nearest Neighbors | 0.770 | 0.701 | 0.795 | 0.696 | 0.616 | 0.730 | 0.927 | 0.940 | 0.329 | 0.284 |
| Neural Network | 0.808 | 0.815 | 0.830 | 0.839 | 0.669 | 0.672 | 0.939 | 0.939 | 0.392 | 0.407 |
| Ridge Regression | 0.854 | 0.878 | 0.879 | 0.911 | 0.697 | 0.677 | 0.947 | 0.945 | 0.485 | 0.556 |
| Random Forest | 0.871 | 0.878 | 0.907 | 0.915 | 0.649 | 0.659 | 0.941 | 0.942 | 0.533 | 0.560 |
| Bayesian Generalized Linear Model | 0.870 | 0.871 | 0.894 | 0.892 | 0.727 | 0.748 | 0.952 | 0.955 | 0.528 | 0.533 |
| Gradient Boosting Machine | 0.874 | 0.883 | 0.910 | 0.918 | 0.657 | 0.669 | 0.942 | 0.944 | 0.544 | 0.574 |
| Bagged Tree | 0.861 | 0.862 | 0.897 | 0.896 | 0.641 | 0.660 | 0.939 | 0.941 | 0.505 | 0.511 |
| Single C5.0 Ruleset | 0.860 | 0.855 | 0.893 | 0.885 | 0.660 | 0.676 | 0.941 | 0.943 | 0.502 | 0.493 |

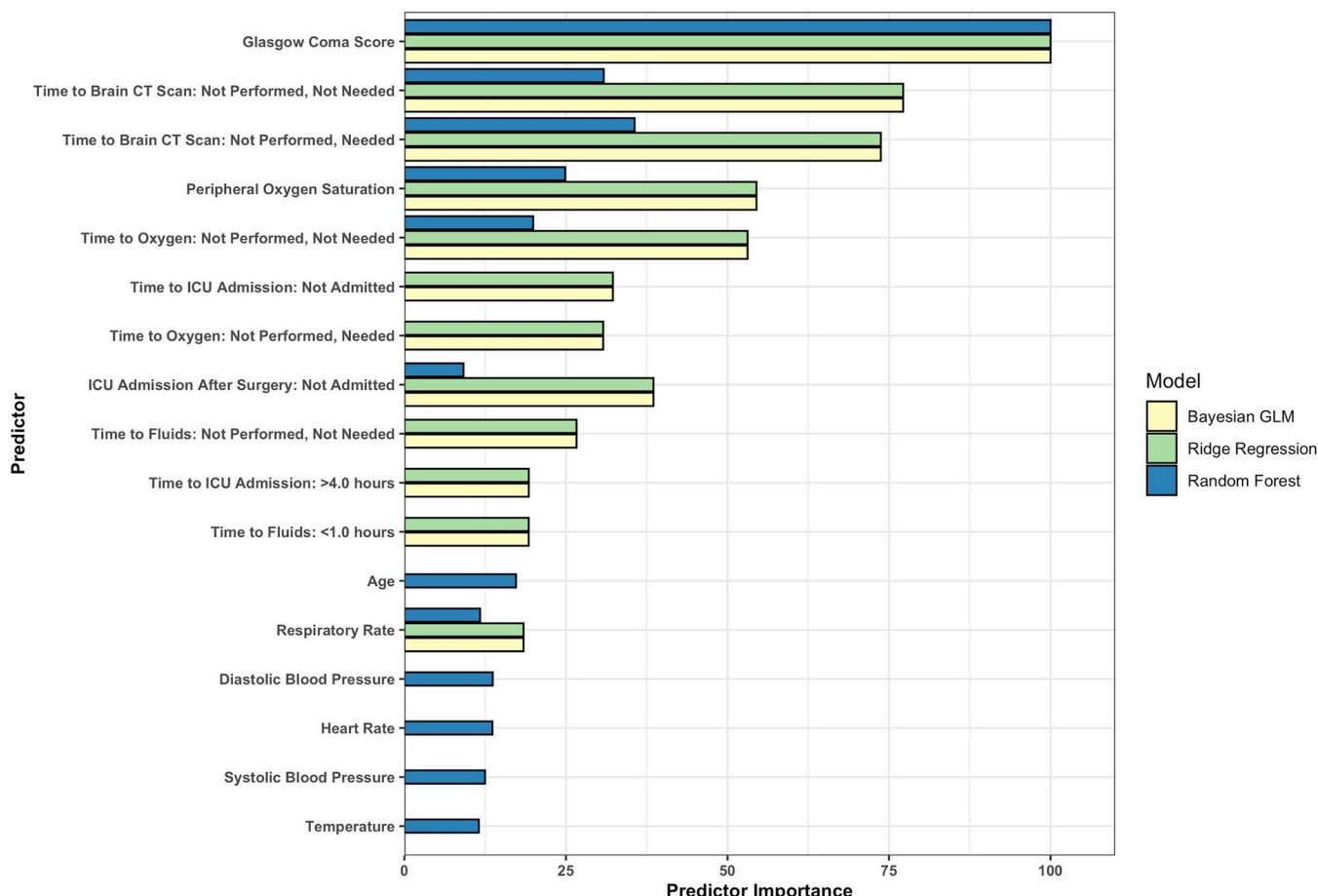

**Fig 2. Twelve most important predictors in the top three performing models.** Predictor importance is scaled to a range of 0 to 100 for easy comparison across all modeling techniques. Higher values indicate greater importance to a model's prediction ability. If a model is not represented by a predictor, it means the predictor was not one of the top twelve most important variables for that model.

models identified only one machine learning based TBI prognostic model, and it achieved an AUC of 89.0 [38]. With an AUC of 89.5, our best performing model exemplifies the value that time to care variables may add to a prognostic model's ability to make TBI outcome predictions.

Second, time to care variables take priority over other model predictors. In our top three performing updated models, time to care variables encompassed many of each model's most important predictors (Fig 2). In other words, time to care variables contributed to model performance more substantially than other clinical and sociodemographic predictors. Notably, the categories "not received, needed" and "not received, not needed" within our time to care variables were often ranked most important to model performance. This suggests that patient need for a specific procedure may be a more powerful predictor of TBI outcome than the length of time a patient spends waiting to receive that procedure after being assessed by a physician. Nonetheless, to support TBI prognostication, ED healthcare providers may benefit from prioritizing the collection of time to care information over other clinical and sociodemographic information when faced with limited resources that may prevent the collection of other more clinically complex indicators that predict TBI outcomes.

## Usefulness and time to care

Clinical prediction models typically provide a patient's risk of some outcome of interest. Our models, for example, predict the risk of a poor TBI outcome. Moreover, the inclusion of time to care variables in our models improves prediction performance which suggests that future efforts to build tools for TBI prognosis may benefit from taking into account information regarding delays to different forms of emergency care. However, physicians agree that knowing just the risk of an outcome is of minimal clinical use [39]. For a prediction model to be more useful in a clinical setting, it should inform whether or not to provide a specific treatment or diagnostic procedure [40]. Our time to care variables include categories that indicate whether or not a patient received lab tests, a chest x-ray, a skull x-ray, a brain CT scan, fluids, oxygen, TBI surgery, non-TBI surgery, and ICU admission. Thus, our prediction models could, theoretically, inform the provision of these procedures. For example, our models could be structured to give a patient's risk of a poor in-hospital outcome under the assumption that the patient receives fluids and the assumption that the patient does not receive fluids. An ED healthcare provider can then compare the risk of a poor in-hospital outcome in each scenario to decide whether or not the patient would benefit from fluids. Few TBI models to date can inform the provision of specific procedures. In Perel et al.'s systematic review, less than 20% of identified TBI prediction models included treatment and diagnostic predictors [36]. Ultimately, our time to care variables could increase the usefulness of TBI prediction models to clinical decision makers not only by improving overall TBI prognosis, but also by informing the provision of specific treatment and diagnostic procedures. However, it must be reiterated that our time to care variables cannot estimate the causal effect of receiving vs. not receiving a specific procedure. These variables simply allow a prognostic model to predict a patient's outcome with and without a specific procedure.

## Usability and time to care

A usable prediction model allows the user to obtain the output with ease and efficiency. In a survey of 137 physicians in the United States two of the most cited limitations to using prognostic tools in clinical practice were poor accessibility and a time consuming and cumbersome process [41]. Moreover, prognostic research in general emphasizes the importance of using clearly defined predictors that can be easily measured so as to maximize usability [42]. While trauma related time stamps are not always easy to record, especially in LMIC settings with limited resources, our time to care predictors are both unambiguous and less complicated to use in comparison to other clinical variables. With any of our time to care predictors the user must only enter whether or not a patient received a procedure and, if so, how long the patient waited to receive the procedure since being evaluated by a physician. Furthermore, time to care information is simple to collect relative to other clinical variables. Receipt of care only requires a record of whether or not a patient received a procedure. A delay to care only requires a record of the time at which an event occurred. Consequently, time to care information is accessible in any context, making it a potentially valuable data source. In addition, if collecting information on all twelve time to care variables proved to be cumbersome, users could also prioritize those variables which we show to be most important to model prediction (ex. time to brain CT scan, time to oxygen, time to ICU admission, and time to fluids).

## TBI outcomes and time to care

The first sixty minutes following sustained trauma has been termed the "golden hour" among emergency medicine providers [43]. The "golden hour" represents a window of time after which the probability of mortality significantly increases in the absence of definitive trauma

care [44]. Although the concept of a "golden hour" has been widely promoted, there is little evidence to suggest trauma morbidity and mortality significantly increase following sixty minutes without management or treatment [44, 45]. Regardless of any evidence that undermines or supports the notion of a "golden hour," it is generally accepted that trauma patients should receive care as soon as possible [46]. The positive impact of our time to care variables on the performance of our prediction models suggests an inverse association between length of in-hospital emergency care delays and good TBI outcomes. Our results therefore provide further support to the notion that delays to appropriate care may significantly impact TBI outcomes.

## Future steps

With the inclusion of new predictors, our TBI prediction models must be externally validated. External validation tests a prediction model's performance on a dataset that was not used for model development, and thus assesses the model's generalizability. External validation is therefore a necessary step towards translating a prediction model into a clinically useful tool [47]. Despite this crucial step, external validations remain unreported for many published TBI prognostic models [36]. To date, the CRASH and IMPACT models remain the most comprehensively validated TBI prognostic models constructed [34, 48, 49]. As our team continues to establish TBI registries in settings outside of Tanzania, we hope to have the data necessary to externally validate our models in a future study. It is also important to note that the most practical prediction model is one that achieves a desired level of performance with the fewest number of predictors. In a future study, with updated data from our TBI registry, we plan to use a feature selection approach to identify whether or not there is a subset of our time to care and or clinical/sociodemographic variables that produces a more parsimonious model, and therefore a more practical model for implementation in resource limited settings.

## Limitations

Although the inclusion of time to care variables improved our model's performance, we must consider the limitations of these variables. First, a patient's time of injury occurrence was self-reported. While our registry includes only patients who sustained a TBI no more than 24 hours prior to hospital arrival, the accuracy of our variable time from injury occurrence to hospital arrival may suffer from recall bias. Second, the registry used in this study only includes patients who sustained a TBI no more than 24 hours prior to hospital arrival. Consequently, the registry likely underrepresents the most severe cases of TBI (i.e. those who died of their injury before reaching the hospital). Given that these patients are expected to benefit the most from timely care, the selection bias inherent in our registry likely underestimates the predicative power of our time to care variables. Third, many patients in our registry did not receive a brain CT scan, fluids, or oxygen. Unfortunately, limitations in our dataset prevent us from knowing exactly why a patient did or did not receive a specific procedure. Having this information would help to both contextualize and generalize our findings. Fourth, we had no way of assessing a patient's need for TBI surgery, non-TBI surgery, or ICU admission and therefore could not assess the value of surgical or ICU need to model performance. Lastly, it is important to highlight that for most models the addition of time to care variables increased model performance metrics by no more than five percentage points. While such an increase indicates the predicative power of time to care variables, the clinical significance of a five-percentage point increase in AUC and other performance metrics can only be assessed through additional studies that measure the costs and benefits of using our models in clinical practice to facilitate TBI prognosis.

## Conclusion

Our study assesses the value of need for care and time to care as predictors of in-hospital outcomes in machine learning based TBI prognostic models. We found that our predictors not only improve model performance, but also comprise a majority of the predictors that are most important to the predictive ability of each model. Given these results, and the simplicity with which need for care and time to care data can be collected, patient needs and patient delays may prove to be an easily accessible and valuable source of information when applying prediction models in low-income settings.

## Author Contributions

**Conceptualization:** Catherine A. Staton, Joao Ricardo Nickenig Vissoci.

**Data curation:** Armand Zimmerman, Cyrus Elahi, Thiago Augusto Hernandes Rocha, Francis Sakita, Blandina T. Mmbaga.

**Formal analysis:** Armand Zimmerman, Cyrus Elahi, Thiago Augusto Hernandes Rocha.

**Funding acquisition:** Catherine A. Staton.

**Methodology:** Armand Zimmerman, Cyrus Elahi, Thiago Augusto Hernandes Rocha.

**Supervision:** Catherine A. Staton, Joao Ricardo Nickenig Vissoci.

**Visualization:** Armand Zimmerman.

**Writing – original draft:** Armand Zimmerman.

**Writing – review & editing:** Armand Zimmerman, Cyrus Elahi, Thiago Augusto Hernandes Rocha, Francis Sakita, Blandina T. Mmbaga, Catherine A. Staton, Joao Ricardo Nickenig Vissoci.

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
