## [Decision Letter · Decision Letter 0]

2 May 2023

PGPH-D-23-00188

Delays to Emergency Care as Predictors of Traumatic Brain Injury Outcomes in Machine Learning Based Prognostic Models for Low-Income Settings

Dear Dr. Vissoci,

Thank you for submitting your manuscript to PLOS Global Public Health. After careful consideration, we feel that it has merit but does not fully meet PLOS Global Public Health’s publication criteria as it currently stands. Therefore, we invite you to submit a revised version of the manuscript that addresses the points raised during the review process.

We look forward to receiving your revised manuscript.

Kind regards,

Bethany Hedt-Gauthier, PhD

Academic Editor

Journal Requirements:

1. Please note that PLOS ONE has specific guidelines on code sharing for submissions in which author-generated code underpins the findings in the manuscript. In these cases, all author-generated code must be made available without restrictions upon publication of the work. Please review our guidelines at https://journals.plos.org/plosone/s/materials-and-software-sharing#loc-sharing-code and ensure that your code is shared in a way that follows best practice and facilitates reproducibility and reuse.

Additional Editor Comments (if provided):

The reviewer comments are very thoughtful and overlap with many of my own. Areas that need particular attention:

1) The addition of these delay variables is not trivial in the real world (per R1) and the final models only have marginal gains in AUC (per R2). This needs more discussion on the true utility of what you are proposing.

2) Many of the most predictive delay variables (per Figure 2) are that items were "not needed" (Brain Scan, Time to Oxygen, Fluids, ICU) or are items Needed and not received (Brain Scan, Oxygen). For the first list, these all have much lower rates of worse outcomes - this isn't a "time to get" but rather a "good enough they don't need". So is this really a delay variable, or some proxy for wellness? For the latter list, is it possible that these are needed but not received because the negative outcome already happened - for example, very sick, we couldn't get the brain scan? So again, this isn't really a delay variable but rather proxy for really sick? Very few of the "important delay predictors" are truly measures of time, but rather measures of not received with needed versus not, which I suspect makes these just proxy measures for very sick versus not. This needs more discussion, particularly on how these get captured and used in both model development and prediction deployment.

3) "Prognostic models are an innovative solution that can help optimize clinical decision making to ensure patients with the greatest need and potential benefit receive care first." This is an interesting framing, given that getting care first could mean delays for others. And delays captured in your model could be results of non-algorithm driven risk assessments. I have a hard time wrapping my head around this chicken-and-egg reasoning, and so ask that authors to add more here.

4) The authors report percent accuracy, sensitivity, etc, but never talk about how they decided what algorithm thresholds for classification for which these measures are reported.

Reviewers' comments:

Reviewer's Responses to Questions

**Comments to the Author**

1. Does this manuscript meet PLOS Global Public Health’s publication criteria? Is the manuscript technically sound, and do the data support the conclusions? The manuscript must describe methodologically and ethically rigorous research with conclusions that are appropriately drawn based on the data presented.

Reviewer #1: Yes

Reviewer #2: Yes

2. Has the statistical analysis been performed appropriately and rigorously?

Reviewer #1: Yes

Reviewer #2: Yes

3. Have the authors made all data underlying the findings in their manuscript fully available (please refer to the Data Availability Statement at the start of the manuscript PDF file)?

Reviewer #1: Yes

Reviewer #2: No

4. Is the manuscript presented in an intelligible fashion and written in standard English?

Reviewer #1: Yes

Reviewer #2: Yes

5. Review Comments to the Author

Reviewer #1: The authors are interested in creating a tool for prognostication for traumatic brain injury (TBI) in low and middle income countries (LMIC’s). They build upon previous TBI prognostication models by adding elements of delay to care. They use data from TBI patients admitted to a single institution in Kenya to derive this model. I thank the authors for addressing a hugely important issue in trauma and global health. TBI is a massive problem worldwide, but especially in LMICs. Any practical tools that can be developed and deployed are helpful. Generally, the concept and methodology for this manuscript are sound. I have only a few comments:

Abstract

- None

Introduction:

- The authors do a good job of explaining the 3 delays. In the description of the reasons for the first delay they ascribe it primarily to an individual’s perceived need. I would add that a big part of it is not just the individuals perceived need but also their perceived ability to reach care that is available, effective and affordable. It is common for an individual to know that they need care but may delay in seeking care because of these other barriers.

- The authors state that healthcare providers can only influence the third delay. This statement ignores the power of research, policy and advocacy work by healthcare providers that is critical for effecting change on the first two delays.

Methods

- Line 134: “which” -> “whom”

Results

- Line 244: this is a minor point but it isn’t a “majority” if it is less than 50%

Discussion

- My main concern with the paper is operationalization of the model. 12 separate time variables seems like a lot of data to be able to collect and use. Now, the authors created multiple models and some of these might have used only some of the variables. But it is difficult to know, from the way the manuscript is currently written, what goes into each model. The details about which variables ended up in each model would be helpful to include in the results and in a figure.

- For particle purposes there is always a tradeoff between the model with the best area under the curve (AUC) and the one that is most user friendly. The authors make some effort to address this in the section on “Usability and Time to Care.” I think the authors could be more specific about which of the 12 variables they recommend. I also think that they are overly optimistic about how easy it will be to collect time stamps on all of these events. Time stamps of events in trauma care are difficult in HICs and can be extremely difficult in settings where resources and record keeping are more difficult.

- The authors present their models but then stop short of making a recommendation. I am most interested to hear which variables are most important for prediction. Therefore, figure 2 is helpful. I am also interested in how good the model is at prediction. Therefore, figure 1 is helpful. But it would be most useful to know the details on the models so these data can be operationalized. It would also be helpful to know if a very simple model (for instance, one that just includes the time variables on time to CT scan, GCS and O2 sat) is almost as good as a model

Reviewer #2: In their study the authors present a TBI outcome predictive algorithm based on machine learning capabilities. Although registry-based data often lack granularity, this limitation seems to have been somehow addressed by the authors, as some supplementary data was extracted in an attempt to evaluate the need for different procedures. This is especially relevant in cases of LMICs where inaccessible or costly yet necessary procedures may be omitted which in turns may influence the certainty in the diagnosis, the treatment of choice, patient outcomes, and in that context, the data on hand. Adding information about the unmet needs for a procedures adds to the credibility of the dataset, which the authors ought to be commended for. The results seem to provide evidence backing up the role of delay parameters in the prediction of outcomes after TBI injuries in lower resource settings. These findings are valuable, especially in the era of growing global health efforts, and may be used to guide future research as well as financial efforts and investments aiming at reducing mortality and morbidity from TBI.

However, some concerns are raised as following:

1) Methods should better describe how TBI was defined? Were patients with severe injury to the skull also considered, where those with so-called “mild-TBI” included? This is essential to assess the external validity of the findings.

2) Did the authors reflect over the possibility of a selection bias since the registry only includes patients presenting within 24-hours from their injury. This would not have been a problem in other study, however, in lower income settings (where delayed presentation is particularly common), and especially when the main aim of the study is to look at delays, such restrictions to the selection of patients warrants careful considerations, as they may limit generalizability of the findings in LMICs. This should hence at least be mentioned in the limitation section of the manuscript.

3) Percentages in Tables 1 and 2 and those described in the text look at the same thing from two totally differing angles which makes it harder to follow what is being referred to. For example, take the time to ED arrival variable. In the text the percentages seem to be expressed as the number of those presenting to the ED at more than 12.0 hours in relation to the total number of patients with either good or poor outcomes – which I personally prefer. The authors mention: “A majority of patients in both the good and poor outcome groups arrived to the ED more than 12.0 hours after injury occurrence (31.6%, 33.3%)” (page 8 line 244). However, in the table the authors calculate the percentages of patients with good/bad outcomes in relation to the total number of patients presenting at more than 12.0 hours (88.2% / 11.8%). I think the author should stick with one way of presenting findings, to avoid conclusion. I would recommend sticking with the percentages used in the text, meaning that the ones in the table may be changed.

4) In addition, it is essential mentioning that although the delay-parameters seemed to improve the accuracy of the predictive model, the AUC ultimately experienced an absolute increase of 5% at best (range: 0.8-5%). I think the authors should spend a bit more time discussing and elaborating on whether the magnitude of this change is in fact clinically relevant and significant. Because, at a first glance, a maximal increase of 5% in accuracy only, may reflect a weak weighing coefficient associated with delay factors, although Figure 2 shows otherwise, especially to unexperienced readers. The authors should hence strive to better explain the relatively low improvements in accuracy after adding delay factors and how such result are able to translate to the findings presented by Figure 2 (or in other words, to the fact that delay factors held such a large share of the contribution in the ML predictive model).

5) In the paragraph usefulness, the authors highlight how their ML-based model can be used out in clinical practice. The authors also give examples as to how that may be the case. However, this whole section depends on the premise of the availability of their ML-algorithm. Is the algorithm available to the public, is there a publicly available app or a domain where it may be tested? If not, the authors may benefit from toning down on the usefulness aspect of their discussion and refrain from alluding to the practicality of their algorithm as a tool to use in the clinics. Consider hence removing the parts from line 348 to 357 of page 13, also since the examples provided are very impractical in a real-life situation. Instead, authors should focus on the valuable insight provided by their study (the fact that delays do play a role in outcomes) and suggest that this information may be taken into account when building actual tools for the early detection of unfavorable outcomes in TBI patients.

6) At the end of their discussion the authors mention: ”We plan to quantify associations between in-hospital emergency care delays and TBI outcomes in a future study”. Could you please explain how such a study would be different from this one? In this study the authors looked at both out- as well as in-hopital delays also, isn’t that right?

7) In the conclusion the authors mention that the predictors were ”novel”. From my understanding of the author’s introduction and methods section, these variables were had previously been described. I hence recommend removing the word ”novel”.

6. PLOS authors have the option to publish the peer review history of their article (what does this mean?). If published, this will include your full peer review and any attached files.

**Do you want your identity to be public for this peer review?** For information about this choice, including consent withdrawal, please see our Privacy Policy.

Reviewer #1: **Yes: **Geoffrey Anderson

Reviewer #2: No

---

## [Decision Letter · Decision Letter 1]

15 Sep 2023

Machine learning models to predict traumatic brain injury outcomes in Tanzania: using delays to emergency care as predictors

PGPH-D-23-00188R1

Dear Dr. Vissoci,

We are pleased to inform you that your manuscript 'Machine learning models to predict traumatic brain injury outcomes in Tanzania: using delays to emergency care as predictors' has been provisionally accepted for publication in PLOS Global Public Health.

Best regards,

Bethany Hedt-Gauthier, PhD

Academic Editor

Reviewer Comments (if any, and for reference):

Reviewer's Responses to Questions

**Comments to the Author**

1. If the authors have adequately addressed your comments raised in a previous round of review and you feel that this manuscript is now acceptable for publication, you may indicate that here to bypass the “Comments to the Author” section, enter your conflict of interest statement in the “Confidential to Editor” section, and submit your "Accept" recommendation.

Reviewer #1: All comments have been addressed

Reviewer #2: All comments have been addressed

2. Does this manuscript meet PLOS Global Public Health’s publication criteria? Is the manuscript technically sound, and do the data support the conclusions? The manuscript must describe methodologically and ethically rigorous research with conclusions that are appropriately drawn based on the data presented.

Reviewer #1: Yes

Reviewer #2: Yes

3. Has the statistical analysis been performed appropriately and rigorously?

Reviewer #1: Yes

Reviewer #2: Yes

4. Have the authors made all data underlying the findings in their manuscript fully available (please refer to the Data Availability Statement at the start of the manuscript PDF file)?

Reviewer #1: Yes

Reviewer #2: Yes

5. Is the manuscript presented in an intelligible fashion and written in standard English?

Reviewer #1: Yes

Reviewer #2: Yes

6. Review Comments to the Author

Reviewer #1: Thank you for addressing my comments.

Reviewer #2: (No Response)

7. PLOS authors have the option to publish the peer review history of their article (what does this mean?). If published, this will include your full peer review and any attached files.

**Do you want your identity to be public for this peer review?** For information about this choice, including consent withdrawal, please see our Privacy Policy.

Reviewer #1: **Yes: **Geoffrey Anderson

Reviewer #2: No
